# Advances in TEE-Centric Intraprocedural Multimodal Image Guidance for Congenital and Structural Heart Disease

**DOI:** 10.3390/diagnostics13182981

**Published:** 2023-09-18

**Authors:** Xinyue Zhang, Jordan Gosnell, Varatharajan Nainamalai, Savannah Page, Sihong Huang, Marcus Haw, Bo Peng, Joseph Vettukattil, Jingfeng Jiang

**Affiliations:** 1School of Computer Science, Southwest Petroleum University, Chengdu 610500, China; xinyuez0703@outlook.com (X.Z.); bopeng@swpu.edu.cn (B.P.); 2Betz Congenital Health Center, Helen DeVos Children’s Hospital, Grand Rapids, MI 49503, USA; jordan.gosnell@helendevoschildrens.org (J.G.); sihong.huang@helendevoschildrens.org (S.H.); marcus.haw@helendevoschildrens.org (M.H.); 3Department of Biomedical Engineering, Michigan Technological University, Houghton, MI 49931, USA; fnvarath@mtu.edu (V.N.); scpage@mtu.edu (S.P.); 4Joint Center for Biocomputing and Digital Health, Health Research Institute and Institute of Computing and Cybernetics, Michigan Technological University, Houghton, MI 49931, USA

**Keywords:** image guidance, structural heart diseases, image fusion, transesophageal echocardiography, medical image analysis

## Abstract

Percutaneous interventions are gaining rapid acceptance in cardiology and revolutionizing the treatment of structural heart disease (SHD). As new percutaneous procedures of SHD are being developed, their associated complexity and anatomical variability demand a high-resolution special understanding for intraprocedural image guidance. During the last decade, three-dimensional (3D) transesophageal echocardiography (TEE) has become one of the most accessed imaging methods for structural interventions. Although 3D-TEE can assess cardiac structures and functions in real-time, its limitations (e.g., limited field of view, image quality at a large depth, etc.) must be addressed for its universal adaptation, as well as to improve the quality of its imaging and interventions. This review aims to present the role of TEE in the intraprocedural guidance of percutaneous structural interventions. We also focus on the current and future developments required in a multimodal image integration process when using TEE to enhance the management of congenital and SHD treatments.

## 1. Introduction

Structural heart disease (SHD) refers to the heart abnormalities that affect the heart’s valves, walls, chambers, or muscles [1]. Its complex etiology includes congenital malformations, molecular or genetic anomalies, rheumatic valve disease [2,3], hypertension, cardiomyopathy, etc. In addition to its impact on quality of life, SHD often leads to severe complications such as heart failure [4], stroke [5], etc. Therefore, the diagnosis and treatment of SHD is a critical area of research in the field of cardiology and cardiac surgery.

The diagnosis of SHD usually requires a variety of imaging tests, including electrocardiography [6], echocardiography [7], computed tomography (CT) [8], magnetic resonance imaging (MRI) [8,9], and invasive cardiac catheterization [10,11]. After diagnosis, SHD is increasingly treated via minimally invasive interventional procedures [12]—e.g., cardiac catheter occlusion or a closure of defects [13], valve replacement or repair, etc.—under imaging guidance [14,15,16,17]. In a well-planned and executed cardiac catheterization procedure, imaging is fundamental for all aspects of the transcatheter treatment process, including pre-procedural planning, the assessment of the extent of the disease, diagnosing and detecting risk factors, and intraoperative guidance, as shown in Figure 1.

Echocardiography is established as an important tool for diagnosing and treating SHD because it can visualize the structure and function of the heart, as well as provide real-time feedback. In particular, transesophageal echocardiography (TEE) is frequently used during the treatment of SHD. TEE places an ultrasound probe through the esophagus to visualize the heart without any interposing structures between the probe and the heart, thus allowing for an excellent acoustic window that enables a clear visualization of the heart with minimal trauma. Furthermore, TEE is a low-cost, real-time, ionizing-radiation-free modality. As a result, under TEE guidance, interventionalists can accurately visualize the moving catheters or devices within the beating heart in real-time.

Although TEE has become an important means of imaging SHD, using other modalities to augment the image guidance offered by TEE is now standard practice. A list of commonly used image modalities can be found in Figure 2. As all medical imaging modalities have strengths and weaknesses, image fusion is widely accepted in the clinical workup of patients with SHD.

The primary goals of this review paper are two-fold. First, this review paper summarizes the state-of-the-art clinical imaging technologies available for managing SHD as well as the limitations that are in need of further development of each modality. Second, our work also emphasizes the continued technological developments toward multimodality fusion that are being conducted to supplement the TEE guidance during treatment of SHD.

The rest of this article is structured as follows. Section 2 discusses the limitations of the single-imaging modality that is frequently used during SHD treatment, which motivates multimodal image integration. In Section 3, the technological developments related to multimodal image integration are introduced, followed by a detailing of the clinical applications centered around TEE-based multimodality integration in Section 4. We close our work by outlining the challenges and opportunities for a multimodal integration in treating patients with SHD. A summary of the key articles reviewed in this paper is provided in Appendix A for the convenience of the readers.

## 2. Strengths and Weaknesses of Single-Modality Intraprocedural Imaging Guidance in the Treatment of SHD

Intraoperative imaging guidance of SHD is used to assist interventionalists and imagers in understanding the catheter and device manipulation required during the procedure [18]. The instantaneous feedback provided through this guidance to the interventionalists is the key to procedural success. Also, providing comprehensive information enables the “interventional team” to avoid potential complications and to modify the surgical plan for better outcomes.

As is well understood in the literature [19], the available intraoperative imaging modalities (i.e., computed tomography, magnetic resonance imaging, X-ray fluoroscopy, and ultrasound) all have their advantages and disadvantages in terms of retrieving the required information, as summarized in the following few subsections.

### 2.1. X-ray Fluoroscopy and Angiography

X-ray fluoroscopy represents the mainstay for intraoperative imaging guidance [20] in patients undergoing many SHD procedures, such as transcatheter aortic valve replacement. Fluoroscopy helps to visualize the blood flow through the arteries and allows for interventionalists to examine vessel patency. Also, fluoroscopy can be used to guide catheter movement/placement, and allows for the accurate positioning/deployment of devices.

The advantages of X-ray fluoroscopy are its wide availability and real-time image guidance. The real-time capability is much needed for catheter movement and device implantation. However, it has significant limitations. First, the soft tissue contrast on fluoroscopy is poor, thus anatomical information is often lacking. The structural information of soft tissues is required to assess the device deployment in most interventions, e.g., transcatheter aortic valve replacement (TAVR). Second, when an X-ray fluoroscope is used for image guidance, patients and interventionalists are exposed to considerable ionizing radiation [21]. The cumulative exposure to ionizing radiation may result in serious health risks, including cancers, and could impact the cognitive function of patients and medical professionals [22,23]. The images obtained through the fluoroscopy of a dynamic three-dimensional structure such as the heart are limited by the planes and angles at which such images are captured.

### 2.2. MRI

In the last two decades, tremendous progress has been made in cardiac MRI. In terms of clinical workflow, cardiac MRI is increasingly used with advanced software and manual annotation so as to comprehensively assess heart structure, function, and hemodynamics [24]. More importantly, cardiac MRI can accurately characterize soft tissues (e.g., myocardium, valve, aorta, etc.) without ionizing radiation [12]. Specifically, MR imaging has multiple sequences—such as black-blood monophasic [25], bright blood imaging, MR angiography (MRA), and phase contrast—which are used to assess cardiac anatomy and function [26]. The black-blood double-inversion recovery (DIR) sequence is useful for imaging myocardium and blood vessels (including the coronary arteries). The use of gadolinium-based agents can shorten T1 relaxation time, thus resulting in a high signal, also known as “hyper-enhancement”, in the T1-weighted imaging areas of myocardial fibrosis. This is due to the increased extracellular compartments and can lead to increased gadolinium retention and delayed washout. MR angiography can be performed with or without contrast to evaluate non-cardiac structures [27]. Phase contrast sequences generate quantitative velocity information for flow velocity and pressure gradient measurements, thus allowing for the quantification of cardiac output, shunt ratio, and valvular regurgitation [28]. Modern cine imaging employs a steady-state free procession (SSFP) technique that allows for the acquisition of a series of images that show cardiac motion, and it can be segmented to measure the volume and ejection fraction. Cine SSFP imaging is the gold standard for assessments of cardiac function. Furthermore, MR imaging technology can provide additional information, such as myocardial fiber orientation and texture, which can aid in diagnosing and treating heart disease.

In short, MR imaging is widely used for pre-treatment assessments of SHD. However, long data acquisition times, the need for anesthesia in young children, and the limited availability of MR-compatible devices limit the use of MRI for intraoperative image guidance in treating SHD.

### 2.3. CT

CT is an imaging technique that uses X-ray scanning to obtain a large number of image slices and uses a computer to combine these slices into three-dimensional image datasets. It plays a fundamental role [29] in the preoperative evaluation of patients undergoing percutaneous and electrophysiological interventions. Modern multi-slice CT (MSCT) has been firmly established as a primary imaging tool in the context of structural heart disease interventions. MSCT and MDCT [30,31] can achieve high in-plane spatial resolution (0.3 mm or less), and the reconstructed slice thickness can be reduced to 0.5–0.6 mm. Furthermore, we can minimize cardiac motion artifacts during the data acquisition process when operating under an ECG-synchronized imaging protocol. MSCT imaging has been used [32] routinely to treat the transcatheter aortic valve in mitral, tricuspid, and pulmonary valve interventions in left atrial appendage (LAA) occlusion and paravalvular leak occlusion.

However, the absorption of X-rays among different soft tissues is comparable; thus, soft-tissue contrast under MSCT is not its strength. Furthermore, the slow frame rate of 3D CT does not allow for it to provide real-time feedback to operators during the treatment of patients with SHD. That is why MSCT has been limited primarily to pre-treatment assessments of SHD interventions. Pre-procedural CT can provide important information such as on the most feasible/optimal access route, accurate measurements of the dimensions, and the optimal viewing angles of fluoroscopy imaging. At the same time, interventionalists depend on live X-ray fluoroscopic imaging for real-time feedback [33].

To address the concerns of ionizing radiation [30], many research efforts have been devoted to minimizing radiation exposure during pediatric CT imaging, including using low-tube-voltage techniques, adjusting tube current to body size, and using geometric dose modulation. Recent developments in CT have also demonstrated that soft-tissue contrast can be improved. Kaneko et al. [34] presented a method for imaging human cardiac structures via X-ray phase-contrast tomography. Compared with traditional X-ray absorption imaging technology, phase-contrast imaging technology has a higher resolution and imaging ability with respect to soft tissue. This method can provide high-quality 3D reconstruction images of human cardiac structures, as well as display fine cardiac structures and details.

### 2.4. Echocardiography

Echocardiography is a non-invasive, real-time, non-ionizing, and contrast agent-free imaging technique [35]. It is widely used for pre-treatment SHD assessments, thereby providing real-time guidance for catheter interventional procedures. TEE imaging is a dynamic, gold-standard imaging method for guiding SHD intervention, allowing for the real-time observation of cardiac motion and blood flow [36,37]. This imaging technique offers high-resolution images of the heart, thus helping doctors to assess the location and extent of lesions, as well as allowing them to choose the best surgical options more accurately.

In catheterization procedures, TEE provides the best spatial resolution and anatomical definition with high image resolutions and a sterile environment that does not interfere with the surgical area, thus making it commonly used in cardiac surgery [38]. It has been used routinely to guide the closure of atrial septal defect (ASD), patent foramen ovale (PFO), and ventricular septal defect (VSD) [39]. Although its use has been shown to reduce fluoroscopy time and radiation exposure, one major limitation of TEE is its small field of view [40] and the need for deep sedation or general anesthesia (GA). Recent reports have found that most patients had esophagus injuries after TEE-guided procedures due to probe manipulation [41]. To a certain extent, frequent probe manipulation is a response to the limited field of view.

TEE has a high risk for contagious infections [42]; thus, prior to the procedure, the risk–benefit ratio in favor of the patient should be paramount. An alternative technique like the LA strain [43] as a predictor of the LA appendage thrombus in the setting of atrial fibrillation is considered by some authors instead of TEE.

In recent years, the role of intracardiac echocardiography (ICE) in SHD interventions has been expanding. Most cardiac structures treated with transcatheter SHD intervention can be imaged using ICE probes that are located at one or more of the five locations—such as the middle of the right atrium (RA), low RA, right ventricular (RV) inflow, RV outflow, and left atrium (LA)—to obtain better views [36]. The effectiveness of ICE in guiding the closure of the percutaneous aortic fistula, aortic balloon valvuloplasty, and evaluating the function of artificial AV has been confirmed in a series of small cases. Studies have reported that percutaneous ASD closure under ICE guidance is safer and better than TEE guidance in shortening fluoroscopy and surgical time, as well as in reducing radiation dose [36]. However, the scarcity of data, introduction of larger sheaths and extra access sites, increased cost, equipment limitations, and steep learning curve limit the use of ICE in SHD. Also, the advantages of ICE over TEE have not been generally accepted yet.

## 3. Review of Existing Multimodal Image Integration Techniques

Each imaging modality has its strengths and weaknesses and may be unable to provide interventionalists with complete information during treatment. Thus, multimodal image integration is a logical solution and has received increasing attention [44].

Without automated multimodal image integration, interventionalists are required to mentally overlay features from multiple modalities together [45] while performing a procedure. Such mental integration can be a daunting task. In contrast, multimodal image integration [46] automatically registers images together and offers excellent visualization. Figure 3 shows a generic process of multimodal image integration starting with the data acquisition of individual modalities, followed by image preprocessing, including image enhancement and denoising, image registration, and feature extraction. At the end of the integration processes, images are displayed for enhanced visualization. Modern computer graphic techniques such as volume rendering, virtual reality [47], and augmented reality have all been used for enhanced visualization.

One of the significant limitations of the current imaging technique is the specialist interpretation of images and the communication between the specialists (i.e., between interventionalist and imager). Automated image integration and advanced visualization remove this interphase through the ‘democratization’ of image interpretation to a common language.

By combining the strengths of each imaging modality, multimodality integration decreases procedure time and presumed radiation dose [37]. Arguably, the amount of contrast agent used might also be reduced.

The major steps related to technological developments in multimodal image integration are summarized in Figure 3 and briefly explained below in this section.

### 3.1. Image Registration

One important step in multimodal image integration is image registration [48], as shown in Figure 4. Mathematically, image registration solves an optimization problem that arises when aiming at maximizing the similarity or minimizing the difference between two images that are acquired from the same subject. Image registration helps ensure spatial consistency between two images that are captured by different imaging modalities because their image formation processes differ vastly.

Medical image registration remains a hot research topic, and existing medical image registration can be divided into two categories: (1) traditional and (2) deep-learning-based registration methods. In the registration process, it is necessary to specify the required registration accuracy and establish a corresponding mapping relationship between a reference image $I_{r}$ and a target image $I_{t}$; the mapping relationship is known as a parameter transformation $T_{g}$, and the mapped $I_{r}$ maximizes its similarity to It via a pre-determined similarity measure. Formally, the goal of image registration is to determine an optimal transformation $T_{g}$ by which transformed $I_{r}$ (i.e., $T_{g}(I_{r})$) can maximize its similarity with the target image It. Formally, the optimization problem is written as follows
(1)$$argmax\;\{S\left(T_{g}\left(I_{r}\right),\;\;\;\;I_{t}\right)+R\},$$
where R is a regularization term, and adding the regularization term R allows for the solution of the transformation $T_{g}$ to be stable.

Among the traditional methods, an array of well-established methods have been defined and widely adopted. Loosely speaking, image registration methods can be performed with variable transformation models and different regularization terms (elastic, diffeomorphic, unbiased, etc.), as well as similarity metrics S(·) (cross-correlation, mutual information, etc.).

Under the category of deep-learning-based image registration methods, information at various spatial scales is extracted and aggregated together to transform the reference image to the target image in four subcategories: (1) deep similarity [49], (2) fully supervised, (3) unsupervised, and (4) dual-supervised methods.

The underlying idea behind deep similarity-based methods centers on using deep-learning-based similarity measures to replace the traditional similarity metrics such as the sum of square distance (SSD), mean squared distance (MSD), (normalized) cross-correlation (CC), (normalized) mutual information (MI), etc. This method has shown its feasibility in multimodal registration between CT and MR [50,51,52]; brain T1 and T2 MRI images [53,54,55]; and MRI and ultrasound [55,56,57,58].

Similar to other applications in deep-learning-based medical image analysis, supervised learning is also mainstream. Those attempts are well documented in the literature. Conceptually, under supervised learning, network models are trained to recognize transformations while minimizing the differences between those transformations and the known ground truth transformations, which is similar to the traditional deformable image registration (DIR) methods. The optimized loss function includes the mean squared error (MSE) and comparative loss so as to minimize the gap between the predicted value and the actual Tg (e.g., Equation (1)). Notably, Eppenhof et al. [59] proposed using synthetic random transformations in order to train a convolution neural network (CNN) to perform 3D CT lung deformable image registration (DIR). Eppenhof et al. [60] also improved the method by using U-net architecture. Both studies demonstrated the possibility of using CNN for direct transformation predictions. However, regarding registering multiple medical images, the ground truth transformations were often unknown.

Considering that, in actual supervised learning registration—due to the difficulty in obtaining real deformation field manually, and because the artificially synthesized label information may introduce unnecessary errors—there is increasing interest in developing unsupervised one-step transformation estimation frameworks [61]. In the unsupervised learning methods of multimodal image registration, known transformations or registration labels are not available as training data. Thus, minimizing the loss functions that measure similarity (e.g., mutual information, normalized cross-correlation, etc.) and consistency (e.g., reciprocal motion consistency) between images are used to guide the training process. The Spatial Transformer Network (STN) [62] technique is one of the earliest methods. It performs spatial operations on data in the network, enabling it to actively perform spatial transformations on feature maps, with the feature maps themselves as conditions. This method is conducted without additional training supervision or modifications to the optimization process. Inspired by STN, more unsupervised registration methods have been developed. As one of the most promising methods of unsupervised learning on complex distributions in recent years, the image registration method based on Generative Adversarial Networks (GANs) has gradually become popular because it can introduce additional regularization, as well as perform image domain transformation so as to convert multimodal image registration to single-modal [63].

Another registration algorithm based on dual-supervised learning involves coupling the label information of samples with a measurement of the similarity between images in the training process. Fan et al. [64] proposed the Brain Image Registration Network (BIRNet), which uses the dual-supervised learning method to obtain the deformation field of 3D brain MRI images. The loss function measures the similarity between the template image and the deformed image to be registered, as well as the similarity between the predicted and actual deformation fields. Cao et al. [65] designed another dual-supervised learning method for cross-modality registration between CT and MRI. Specifically, using the aligned CT and MRI image pairs, the multimodal registration problem was transformed into a single-mode image registration problem, and similarity measures under a single modality were used instead of similarity measures between two modalities. In the loss function, the differences between CT-CT images and the similarities between MRI-MRI images were calculated. In general, compared with unsupervised learning, dual-supervised learning makes full use of labeled sample data information, which can make up for the shortcomings of unsupervised learning to a certain extent but still cannot avoid the restriction of using labeled information.

### 3.2. Image Registration

After multimodality images are spatially registered, we must combine the information with varying fusion technologies. The image fusion technologies in question can be further divided into three categories: (1) pixel-level fusion, (2) feature-level fusion, and (3) decision-level fusion. Pixel-level fusion is achieved by obtaining a fused image through directly processing the data acquired. This method requires a perfect registration of the source images to combine each pixel, but it is the most computationally efficient of the three categories [66]; it is also the basis of high-level image fusion and one of the focuses of current image fusion research. Due to its ability to preserve spatial details in fused images, this method is more suitable for medical imaging applications.

In contrast, feature-level fusion algorithms extract features of the source image and fuse them feature by feature. As a result, feature-level fusion may be more robust to noise and less sensitive to inconsistencies between two different modalities. Finally, decision-level fusion algorithms [67] allow for an effective combination of information at the highest level of abstraction, directly combining image descriptions and making decisions based on the ultimate purpose of the application.

Multimodal image integration in the context of clinical workflow (i.e., multimodal medical image fusion) is largely based on pixel-level fusion. Thus, our review below focuses on several mainstream fusion methods (as shown in Figure 5): spatial domain, transform domain, fuzzy logic, sparse representation fusion, and deep learning.

#### 3.2.1. Spatial Domain Fusion

Historically, spatial domain-based medical image fusion technology was of high interest. Its fusion methods were simple, and the fusion rules could be directly applied to the source image pixels so as to obtain the fused image. The generic fusion methods in the spatial domain can be as follows:(2)$$I_{F}=F(I_{1},I_{2},\ldots ,I_{K}),$$
where F(·) represents a “fusion operator”. Depending on the specific fusion operator selected, fusion methods include intensity-hue-saturation (IHS) transformation, high-pass filtering, principal component analysis (PCA), averaging, maximum selection, minimum selection, and Brovey transformation (BT) [68].

The remote-sensing research community widely uses the IHS image fusion technique to fuse multispectral (MS) and panchromatic (PAN) images. On the one hand, a panchromatic image is a high-resolution image with no colors, which shows a scene to the human eye (acquired using a single wavelength band). On the other hand, a multispectral image is a low-resolution image with many layers of the same scene that possess rich spectral information. Each layer is acquired at a particular wavelength band (for example, a red–green–blue channel image) [69]. Fusing MS and PAN images produces a combined image with a high spatial resolution and richer spectral information [70].

Principal component analysis (PCA) projects the data to its eigenspace. First, the images are divided into many matrix blocks. Second, for each matrix block, the principal eigenvector is calculated. Suppose $x^{T}$ and $y^{T}$ are eigenvectors of images $I_{A}$ and $I_{B}$, respectively. Finally, fusion can be performed based on the rule $IF=\omega 1\times I_{A}+\omega 2\times I_{B}$, where ω1 and ω2 are weights based on x and y [71].

The Brovey (color normalization) transformation (BT) can be used to fuse the PAN and MS images. The BT preserves the high degree of spectral information of MS images, as well as the spatial information of PAN images. Each layer in the color of the MS image is multiplied by a ratio of the high-resolution PAN image data that has been divided by the sum of the color MS band images [72].

However, due to the spectral and spatial distortions in the fused image of the spatial domain, the popularity of this research area has declined in recent years. Researchers have started using spatial domain fusion strategies as a component of transformation domain methods to develop new research methods [67].

#### 3.2.2. Transform Domain Fusion

Unlike the spatial domain fusion method, which directly processes image pixel values, the transform domain fusion method first converts the input image to a particular transform domain (e.g., frequency domain, wavelet domain, etc.). Then, it performs fusion operations in the transform domain. Finally, it transforms the fused representation to the spatial domain so as to obtain the fused image for visualization. The transformed domain-based methods can be summarized as follows,
(3)$$I_{F}=T^{-1}(F(T(I_{1}),T(I_{2}),\ldots ,T(I_{K}))),$$
where $T\left(\cdot \right)$ represents a multiscale transformation, the superscript −1 denotes an inverse operation, and F(·) is the applied fusion operator. Many fusion algorithms adopt multiscale methods to reduce the algorithm’s sensitivity to changes in the grayscale values of the source image, thereby improving the stability and effectiveness of the fusion process. Two classical transform domain fusion methods are briefly described below.

Laplacian pyramid.

The traditional Laplacian pyramid-based fusion algorithm [73] can retain the edge details well, as well as make full use of the information at different scales. It represents an image as a series of quasi-bandpassed images, and it is localized in space and in a spatial frequency. Laplacian pyramid coding involves constructing a Gaussian pyramid, obtaining Laplacian pyramid images as the difference between successive Gaussian levels, quantizing them to yield the compressed code, and reconstructing the image via the expand-and-sum procedure. The Laplacian pyramid encoding scheme is computationally simple, local, and parallelizable, making it suitable for real-time processing. Based on this, Du proposed a method combining the Laplacian pyramid with multiple features to accurately transfer the salient features from the input of medical images to a single fused image, thus solving the problem of limited image contours and contrast representation [74]. They used the Laplace pyramid to convert the input image into its multiscale representation, as well as extracted the contrast and contour feature maps from each scale of the image. This effective fusion scheme combines pyramid coefficients and obtains fused images through the reconstruction process of the inverted pyramid [74].

Multiscale decomposition.

The method based on multiscale transformation is the most popular pixel-level fusion algorithm. The medical image fusion methods of the transform domain are mainly based on the multiscale transform (MST) theory, and they have also been the hotspots of research in recent years. Fusion methods are based on multi-resolution analysis, such as the discrete wavelet transform (DWT) [18,75] and the stationary wavelet transform (SWT) [10]. The MST-based fusion method is divided into decomposition, fusion, and reconstruction. The source images are first decomposed into low- and high-frequency sub-bands via the appropriate MST method. Then, a fusion rule is applied to combine the sub-band coefficients of each source image. Finally, an inverse MST is performed to obtain the fused image [66].

Below, we discuss the three most commonly used transformations in medical image fusion methods: non-subsampled contourlet transform (NSCT), non-subsampled shearlet transform (NSST), and discrete wavelet transform (DWT).

Nonsubsampled contourlet transform.

Figure 6 shows the decomposition process of NSCT. NSCT is a shift-invariant, multiscale transform that decomposes the images based on the nonsubsampled pyramid (NSP) structure and nonsubsampled directional filter bank (NSDFB). The NSP decomposes the image into low-pass and bandpass-filtered images. Then, the bandpass images enter the NSDFB and are decomposed directionally into the sub-bands’ 2D frequency domain. This method has been proven to effectively fuse CT and MRI scans [66].

2.Nonsubsampled shearlet transformation.

Non-subsampled shearlet transformation (NSST) is a non-subsampled multiscale transformation method based on the theory of shear wave transformation. NSST decomposes images into multiscale, multidirectional images through multiscale and multidirectional decomposition. First, an NSP is used as a multiscale decomposition filter to decompose the image into a low-frequency sub-band and a high-frequency sub-band. The high-frequency sub-bands are then decomposed using a shear filter (SF) to obtain multidirectional sub-bands. The NSST decomposition process does not sub-sample the NSP and SF, so the NSST is shift invariant.

3.Discrete wavelet transformation.

The discrete wavelet transformation (DWT) decomposes the image into four high- and low-frequency sub-bands by using a finite set of wavelets [76] that contain both spatial and temporal information (see Figure 7). Then, the source image bands are typically combined via averaging. Specifically, wavelet domain fusion methods use wavelet transformation to decompose an input image into sub-images of different scales and orientations. Then, fusion operations, such as weighted averaging, selection of maximum absolute value, etc., are performed on each sub-image. Finally, the fused representation is transformed back to the spatial domain by an inverse wavelet transform approach. This method has limitations as it cannot capture the smoothness of the images and can have artifacts along the contour edges [66].

#### 3.2.3. Fuzzy Logic Fusion

Fuzzy Logic Fusion [77] is an information fusion method based on fuzzy logic. Fuzzy logic is a mathematical framework handling information with uncertainty. It deals with this uncertain information by introducing fuzzy sets and fuzzy operations. The advantage of fuzzy logic fusion in medical image fusion lies in its ability to deal with uncertain, fuzzy, and incomplete information. Compared with traditional pixel-level fusion methods, fuzzy logic fusion can better preserve the information of the input image and reduce artifacts and distortions. In addition, the fuzzy logic fusion method has good interpretability and scalability, and it can be adjusted according to different application scenarios and requirements.

Fuzzy logic can be used in conjunction with other pixel-level techniques to enhance image contrast. Fuzzy logic uses a fuzzifier, an interference engine, a defuzzifier, fuzzy sets, and fuzzy rules to change the membership functions that determine the variable weight of each pixel. Javed et al. used wavelet decomposition and fuzzy logic with backpropagation and least mean square algorithms to successfully fuse MRI and PET scans of a higher quality than previous methods [78].

#### 3.2.4. Sparse Representation Fusion

Sparse Representation Fusion is a multimodal technology based on sparse representation theory, and it is widely used [79] in medical image fusion. Sparse representation theory [80] states that a signal can be decomposed into a linear combination of a set of sparse vectors. This method first decomposes the image into sparse representations, combines these representations according to a specific fusion strategy, and then finally reconstructs the fused image. The advantage of sparse representation fusion in medical image fusion lies in its ability to effectively extract and preserve image feature information in the sparse domain. Through an appropriate fusion strategy, the complementarity of different image features can be achieved, thereby improving the quality of the fusion images. Yang was the first to apply coefficient theory to image fusion. He proposed an algorithm [81] derived from a sparse decomposition of multi-source images in the same dictionary. Specifically, the sparse coefficient calculated at the same (anatomical) position of multiple images is taken as the absolute value of the larger value to obtain the fused sparse coefficient, which is then combined with the dictionary to reconstruct the fused image.

The standard sparse representation method does not consider the inherent structure of sparse signals, i.e., non-zero elements that appear in clusters or groups (this structure is called group sparsity). To address this problem, Li et al. proposed a novel dictionary-learning method called Dictionary Learning with Group Sparsity and Graph Regularization (DL-GSGR) [82]. This method ensures that the learned dictionary has low group coherence by alternating group sparse coding and dictionary updating, thus enabling the effective group sparse coding of any signal. In addition, the method uses graph regularization to model the geometric structure of atoms.

The limitations of fusion methods in this category are also noted in the literature. First, sparse representation fusion involves computationally intensive tasks such as dictionary learning, sparse coding, and reconstruction. Sparse representation fusion requires solving the L1 norm optimization problem, which leads to high computational complexity, thus increasing the computation time. Second, the performance of sparse representation fusion depends strongly on the learned dictionary. If the dictionary does not capture the image features well, the fusion may not be satisfactory.

#### 3.2.5. Deep Learning Fusion

The medical image fusion methods in the spatial and transform domains resulted in a low correlation between the original and fused images [83]. Deep learning methods have received rapid attention in the last decade. Compared with spatial domain and transform domain-based methods, deep learning methods have stronger adaptivity, robustness, and accuracy; in addition, they can complete the task in a shorter time and improve the quality of medical images. Therefore, deep learning methods have great potential in medical image fusion. Research on several common models, such as CNN, Restricted Boltzmann Machine (RBM), AutoEncoder (AE), and U-net, has been conducted in the field of medical image fusion. Among them, CNN is the most widely used. A general image fusion framework based on CNN is shown in Figure 8.

CNN is a typical deep learning model [84] that attempts to learn hierarchical feature representation mechanisms for signal/image data with different levels of abstraction. Performing activity level measurement and fusion rules is often difficult in existing image fusion methods. Through using a learning CNN, a multi-focus image fusion method [74] based on deep learning was proposed to solve this problem by directly mapping the relationship between the source and fused images.

In recent years, with the popularity of transformers, many researchers have also applied CNN in image fusion. Early methods applied a CNN-based multi-focus algorithm, creating a decision map based on the image focus that determines how each pixel is weighted [85]. However, this method works only in the spatial domain and requires that the source images have similar local structures. Unfortunately, this requirement cannot be satisfied in multimodal image fusion under many circumstances. Liu et al. addressed this problem by decomposing the images via a Laplacian pyramid and a weight map [86] instead of direct coefficients so as to preserve source image detail. A combination of CNN and sparse autoencoders (SAE) was used to fuse multimodal neural image data from MRI and FDG-PET [87], thus achieving significant performance improvement in both directions compared with the single modality method. Zhao et al. proposed a novel Correlation-Driven Decomposition Fusion Network (CDDFuse) to address the challenges of cross-modal feature modeling and expected feature decomposition. The network employs Restormer blocks to extract shallow cross-modal features, a dual-branch Transformer CNN feature extractor with Lite Transformer (LT) blocks for processing low-frequency global features, and Invertible Neural Network (INN) blocks for extracting high-frequency local information. With a correlation-driven loss, CDDFuse can make the low-frequency features correlate while keeping high-frequency components uncorrelated, thus further improving image fusion performance. Experiments showed promising results for CDDFuses in various fusion tasks, as well as enhanced performance for downstream infrared-visible semantic segmentation and object detection.

## 4. Current Clinical Applications of TEE^+^ for Intraprocedural Image Guidance

In 1972, two-dimensional TEE was first used to evaluate the results of mitral valve surgery [88]. The development of matrix array transducers (50 rows by 50 columns) enabled real-time 3D echocardiographic data acquisition. Currently, major ultrasound vendors offer 3D/4D TEE through their matrix array transducers (Z6M True Volume 3D TEE Probe, Siemens Healthcare, Mountainview, CA, USA; X7-2t, Philips Healthcare, Andover, MA, USA, 6VT-D, GE Healthcare, Wauwatosa, WI, USA). As a result, modern TEE provides useful real-time three-dimensional morphologic assessment and hemodynamic interrogation. Technological developments, including multimodal integration with other modalities, have expanded TEE’s clinical applications for examining cardiac function, structures, and hemodynamics [89].

Recent clinical guidelines indicate the importance of echocardiography in the perioperative evaluation of heart disease. For example, TEE images are used to confirm the position and size of perivalvar leaks, as well as to obtain measurements of the aortic annulus and aorta during surgery, evaluate the intraprocedural progress of aortic balloon valvuloplasty (ABV), and assist in locating the catheter and valve position during percutaneous transcatheter heart valve (THV) deployment [90]. TEE guidance is also used in the interventional management of congenital heart defects like ASD and VSD closure, in the surgical repair of the aorta and aortic dissections, as well as in other applications such as valve repair, cardiopulmonary transplantation, the insertion and monitoring of ventricular assist devices, the assessment of the site and severity of paravalvular regurgitation [91], etc.

However, the field of view of TEE transducers is still relatively limited, making it challenging to examine the entire region of interest without moving the transducer. In many centers, fluoroscopy and TEE or cardiac CT are displayed on different screens in different orientations during an SHD procedure. Without automated live multimodal integration, interventionalists must mentally integrate multimodal images to understand the anatomy and spatial relationships during the procedure. Mental image integration may become problematic during communication amongst clinicians when mentally integrated images differ from person to person. In contrast, fused images improve eye–hand coordination as multimodal imaging information is overlaid. Furthermore, virtual markers can be added to fused images to improve visual coordination. These virtual markers in live integration allow for the democratization of multimodal imaging, in which multiple clinicians can view the same region of interest created by fused medical imaging, which may minimize misunderstandings, especially in cases involving complex or atypical anatomy.

### 4.1. Fusion Imaging of 3D TEE with Fluoroscopy

TEE-fluoroscopy fusion has been tested on many SHD interventions in the clinical setting. Recall that TEE’s limitations include operator dependency, low image quality, and poor temporal resolution (if 3D data acquisition is used). Furthermore, TEE imaging alone can have a difficult time visualizing cardiac structures due to acoustic artifacts (e.g., prosthetic intracardiac devices and eructation), thus prohibiting imaging windows that may be essential to view specific regions of interest, such as identifying the location of paravalvular leaks after a prosthetic valve implantation.

Two integrated (TEE/fluoroscopy) systems are commercially available in the clinical workflow. The EchoNavigator software (Philips Healthcare, Best, The Netherlands) performs real-time 3D TEE and fluoroscopy fusion on a single screen. EchoNavigator starts with TEE probe localization and calibration. More specifically, EchoNavigator software [92] automatically finds and tracks the TEE probe’s position and direction within the fluoroscopic image. Post-co-registration, the EchoNavigator system automatically aligns the two images and compensates for any change of position in the C-arm [69]. In 2017, TrueFusion™ fusion-imaging technology (Siemens Healthineers, Erlangen, Germany) also received US FDA clearance for providing image guidance during SHD procedures. In a TrueFusion system, fluoroscopy images are registered with echo data by machine learning algorithm-based probe detection. Then, automated registration can integrate the two modalities. In both systems, markers labeled on echography are automatically transferred to the registered fluoroscopic data.

There are also some TEE-fluoroscopy prototype research systems. One approach [72] involves automatically relocating the 3D TEE image whenever the C arm of fluoroscopy moves by using image-based tracking and allowing the echocardiographer to manipulate the image through using a table-sided mouse. A discriminative learning-based method [90], which generates the required training data by automatically generating a single-volume TEE image, has been proposed to adapt to actual X-ray data, thereby avoiding manual labeling costs. This method uses unlabeled perspective images to estimate the differences in feature space density, and it allows for the correction of covariate bias through instance weighting so as to fuse image data from transesophageal ultrasound (TEE) and X-ray perspective imaging for use in the minimally invasive treatment of SHD. The evaluation shows that, compared with cross-validation on the test set, this method reduces the detection of faults by 95% and the positioning error from 1.5 mm to 0.8 mm [88,89,90,91,92,93].

### 4.2. Clinical Applications of TEE^+^ Fusion Imaging

#### 4.2.1. Transseptal Puncture

Transseptal puncture (TSP) is the most common transcatheter intervention for treating SHD. It is an interventional technique where a needle is used to puncture the fossa ovalis to access the left atrium. The procedure occurs typically in the initial stage of more complicated procedures such as left atrial appendage occlusion or percutaneous mitral valve repair/replacement. Several studies [94,95,96] have shown that combining echocardiographic images with X-ray fluoroscopy images in real time can aid in guiding the puncture procedure. Figure 9 shows that echocardiography provides excellent visualization of transeptal puncture.

In the literature, multiple reports found that using integrated TEE/fluoroscopy (i.e., only EchoNavigator) significantly decreased procedure time [97]. However, some studies reported reduced radiation dosage, while other studies found the radiation dose did not change [98,99].

#### 4.2.2. Left Atrial Appendage Occlusion

LAA occlusion, in which a device is implanted percutaneously to occlude the mouth of a left atrial appendage, is a procedure that requires multimodal imaging for planning and guidance. For these procedures, a live multimodal imaging system could be useful during three key stages: (1) the selection of a transseptal puncture site and subsequent guidance, (2) anatomical orientation and device deployment, and (3) post-deployment evaluation to ensure adequate sealing and device stability. The use of TEE-fluoroscopy systems for transseptal puncture was described in the previous section. After completing the transseptal puncture, there are two other major steps: transseptal crossing of the guiding catheter into the left atrium and the deployment of the occlusion device into the LAA. Some interventionalists perform these two steps using fluoroscopy alone. However, TEE-fluoroscopy integration offers additional benefits, as shown in Figure 10. Furthermore, 2D TEE can measure the orifice of the appendage in different 2D planes, while 3D TEE can more accurately assess the appendage’s perimeter and landing zone (see Figure 11). As a result, the device size can be better selected [100], and visualizing the LAA landing zone allows the device to be appropriately implanted. 

TrueFusion technology has been found to significantly improve procedure efficiency and reduce contrast use [37]. Similar findings were reported by Jungen et al. when using the EchoNavigator system [37].

#### 4.2.3. Mitral Valve Repair

During the procedure, 3D-TEE is often used to evaluate the morphology and function changes of the mitral valve, including the leaflets, annulus, and subvalvular apparatus. TEE can also be used to evaluate the effectiveness of mitral valve repair [101]. However, the orientation of the guiding system and the dedicated structures of valve implants (e.g., MitraClip^®^) are less visible through TEE. The fluoroscopy system serves implant tracking considerably better. Therefore, a multimodality approach for the guidance of MitraClip^®^ implantations when using 2D/3D TEE and fluoroscopy is important.

The EchoNavigator software was shown to reduce the radiation dose and treatment time for MitraClip procedures [102]. As shown in Figure 12 and Figure 13, TEE can already provide excellent information. Visualization can be further improved with the fusion imaging provided to interventionalists. Furthermore, the guidance provided by virtual marker placement on TEE images, with automatic transfer to fluoroscopic images, can help prevent injury to the left atrium and atrial septum during clip implantation [103]. It has also been demonstrated that complex interventions with multiple clips have an increasingly higher reduction in radiation and treatment time when using TEE-fluoroscopy fusion software.

#### 4.2.4. Atrial Septal Defect Occlusion

In the case of the percutaneous closure of atrial septal defect (ASD), TEE, i.e., real-time 3D-TEE, has now been widely established as a primary modality for intraprocedural guidance for ASD occlusion. In leveraging the availability of miniaturized matrix array transducers, 3D-TEE can offer image guidance in real time (see Figure 14), and it allows for accurate measurements of the ASD (i.e., morphology, size, position, and adequacy of the rims around the ASD defect), along with device visualization during the procedure and in post-deployment. The ASD measurements derived from 3D-TEE imaging are essential for determining which device(s) are suitable for ASD closure [104]. Because it is a real-time modality, 3D-TEE can also be used to accurately localize guidewires and catheters [103] during the procedure, e.g., visualizing the wires crossing the ASD and the device’s deployment (see Figure 14). Real-time 3D guidance has significantly reduced the procedure time by removing the need for balloon sizing and reducing the radiation dose.

#### 4.2.5. Paravalvular Leak Closure

Paravalvular leak (PVL) is a known surgical and transcatheter valve replacement complication. Patients most commonly present with congestive heart failure and/or hemolysis, and thus historically require repeat surgery to correct the PVL with mixed success. Consequently, percutaneous PVL closure methods have been developed, and considerable emphasis has been placed on multimodal imaging for diagnosis and surgical guidance in cardiac catheterization laboratories. Several large series of percutaneous PVL closures have been published recently, with encouraging results regarding procedural success and clinical outcomes.

Real-time CT-fluoroscopy fusion has been attempted in PVL closures, especially in apical surgery [105]. More recently, EchoNavigator was used to assist interventionalists for treating PVL [106,107]. Ahmed et al. particularly noted that EchoNavigator helped identify the exact site, size, depth, and shape of the PVL, and it subsequently facilitated navigating a guidewire across the leak [107]. TEE imaging can be useful in PVL cases, as shown in Figure 15; however, imaging may be further limited because the presence of the prosthetic valves creates acoustic artifacts (i.e., acoustic shadowing and reverberations). Thus, TEE image quality may be significantly affected [108].

#### 4.2.6. Transcatheter Aortic Valve Replacement

Transcatheter aortic valve replacement (TAVR) is a valuable alternative for patients who are not good candidates for surgery (e.g., patients with severe, symptomatic aortic stenosis). Interventionalists require essential information about the size and morphology of the aortic root and its surrounding structures to plan the TAVR [109]. Under the integrated TEE/fluoroscopy system, TEE is often used to identify significant landmarks of the aortic root, the aortic cusps, etc., which are invisible on fluoroscopic images. The device visualization (e.g., the orientation of the prosthetic valve) is also critically important but often conducted via the fluoroscopic system.

Under a TEE/fluoroscopy system, interventionalists can flag anatomical landmarks in the echocardiography system, and those virtual markers can be transferred onto the fluoroscopic system. For example, interventionalists can first mark points through the annulus of the valve via the TEE (see Figure 16 and Figure 17), and they can then find additional points that are orthogonal to the annulus (i.e., finding a longitudinal direction of the prosthetic valve when using fluoroscopy). The availability of the longitudinal direction facilitates the device deployment.

## 5. Discussion

Percutaneous structural heart technologies may enable the treatment of patients with minimally invasive approaches. Right now, those percutaneous technologies enable the treating of SHD patients with a high surgical risk. The increasing use of percutaneous structural heart technologies demands a robust approach to intraprocedural guidance; one that provides detailed, real-time 3D spatial orientation and a definition of cardiac structures and defects. That is why augmenting intraprocedural imaging must be one of the priorities when starting or growing a structural heart program.

Although intraprocedural 3D TEE guidance offers the best (overall) solution, at least in the SHD arena, it has not been accepted by all centers as a standard of care. The limitations of TEE (e.g., small field of view) can be mitigated by multimodal image integration. Clinically, evaluating the effectiveness of TEE-centric multimodal integration is determined by the following metrics: (1) increased procedural success rate, (2) reduced procedure time, (3) lower radiation exposure, (4) reduced complication rates, and (5) the need for lesser contrast media. According to TEE guidelines by the American Society of Echocardiography, the Society of Cardiovascular Anesthesiology, and the Society of Thoracic surgeons, TEE adds value to surgical decision making in the operating room, and it has contributed to improved outcomes during the treatment of patients with SHD [110].

Advances in imaging technology have begun to allow for the fusion of multiple imaging modalities. Reliance on a single modality is not always appropriate for cardiac interventions. As demonstrated by TEE-fluoroscopy and CT-fluoroscopy fusion software, multimodal imaging provides significant benefits by essentially merging the strengths of separate imaging modalities. Fusion imaging may provide the most useful diagnostic information for the interventionalist and the surgeon, consequently allowing the patients to receive the best care possible. Recent research has shown that the fusion of real-time TEE with fluoroscopy enhances cardiac interventions and catheter guidance. In contrast, TEE and CT fusion advance SHD diagnoses and interventional device sizing. TEE-fluoroscopy fusion has many advantages, as shown by the lowered procedure times and radiation doses in mitral valve repair and left atrial appendage occlusion (LAAO). Fluoroscopy and CT are the only other image scans being fused with TEE in a clinical setting.

While the current TEE-based multimodal imaging (i.e., EchoNavigator) is an excellent beginning, there are still limitations. The primary limitations of TEE-fluoroscopy imaging are two-fold. First, merging 3D TEE imaging volumes on a 2D angiographic plane is counter to the goal of multimodal imaging since it detracts from the utility of the 3D TEE imaging volumes. Second, the visualization of 3D imaging volumes on a 2D screen inhibits effective 3D spatial navigation of the anatomy. To realize immersive multi-modal 3D imaging, 3D volumes of different imaging modalities will need to be fused and then visualized on an augmented or mixed reality platform. In the multimodal image integration setting, if each modality’s strength is independently realized and combined with an augmentation of characteristics that are preferred for that specific intervention, then multimodal image integration would have come to age.

Unfortunately, some controversies remain regarding the TEE+ fusion strategy. Several studies have shown that TEE+ fusion leads to shorter procedural duration, reduced radiation exposure, and improved patient outcomes. However, due to the lack of standardization, the discrepancies in those studies are difficult to reconcile. For instance, the procedure time was a variable in several studies, and a possible explanation could be that the fusion technology may be more helpful for novice teams in facilitating the learning curve of the procedures. Also, how such a TEE+ fusion system impacts routine clinical practice and patient outcomes compared with the current standard of care is not entirely clear. Furthermore, the current system enhances the use of anatomical information only. In other words, function information (e.g., hemodynamics and soft tissue information) has not been added so that the treatment of SHD can be improved. For instance, using the hemodynamic information available from TEE may enable interventionalists to eliminate the need to use pulmonary artery catheters, thus reducing patient safety risks during the procedure. Ultrasound-based elastographic information can also be used to assess the biomechanical properties of the myocardium, which could be important for some surgical SHD procedures, such as valve repair. We envision that developing a TEE+ fusion system that offers combined anatomical, functional, and automated ultrasound probe manipulation is a worthy direction to pursue.

We anticipate the research to propel the adaptation of artificial intelligence (AI) in the three major components of multimodal image integration (segmentation, registration, and fusion). Augmented and virtual reality following multimodal image integration will likely elevate the use of medical imaging for treating patients with SHD. Such an AI-powered software platform (from multimodal image integration to advanced visualization) could also be foundational in training future medical professionals. AI-guided visualization and automated skill assessments built into our system may accelerate clinicians’ transition from novice to expert. Success stories in TEE training will be exponential if appropriate 3D visualization-based training on an integrated platform can be established [111,112].

Another imaging technique that aids in the planning and guidance of interventions for SHD is 3D printing (3Dp). The benefits of point-of-care 3Dp anatomic models have become increasingly recognized over the past decade. Patient-specific anatomic models allow for a unique, tactile understanding of anatomic features, especially in the context of complex congenital heart disease. Furthermore, 3Dp also allows for ex vivo simulations of transcatheter device deployment, providing another level of assurance during the interventional planning of atypical cases. The maturity and accessibility of 3D printing technology, as well as the application of virtual and augmented reality technology, can help physicians better understand the patient’s cardiac anatomic relationships, thus providing unique opportunities to improve surgical training and pre-operative planning. However, the time and cost required to create models, the difficulty in segmenting finer cardiac structures such as cardiac valves, and the inability to reconstruct dynamic motion limits the overall utility of 3D printing for cardiac cases [113]. However, AI-guided segmentation could assist in overcoming some of these obstacles.

The latest research shows that extended reality (XR) technology [79] Is expected to be an alternative to other 3D technologies in assisting in cardiac interventional planning and guidance. This technology has the potential to offer an intuitive and immersive experience in a three-dimensional space, enabling a thorough understanding of complex spatial relationships when planning and guiding cardiac procedures for congenital and structural heart diseases. This technology surpasses traditional 2D and 3D image displays, but must overcome several challenges to be applied safely and effectively in the clinical space.

As reviewed in this paper, multimodal image integration has advanced in the past decade, from inaccurate spatial transformations to highly complex deep learning algorithms. Now, commercially available packages (e.g., EchoNavigator (Philip]) and TrueFusion (Siemen])) can be used in the clinical workflow; moreover, augmented intraoperative image guidance is now accessible for SHD interventions. A limited number of SHD interventions are performed using the multimodal image fusion of TEE and another imaging technique. With the broad spectrum of available SHD treatments and the small amount of research on multimodal TEE-based guidance, there is still room for innovations in this field. Notably, it would be interesting to fuse MR images with TEE scans, as MRI produces a much-improved characterization of soft tissues or uses ultrasound-based tissue characterization to augment conventional TEE scans.

Furthermore, complex interventions needing image fusion may be more time consuming than a simple procedure that may not need image fusion. A simple ASD closure, for example, does not need image fusion and could be performed in a few minutes, whereas a complex ASD morphology may take much longer, even with multimodal image integration.

There is enormous potential for AI solutions in interventional cardiology. One such area is deep-learning-based image fusion. For instance, current state-of-the-art multimodal image integration is still at the pixel level and primarily limited by two complementary modalities. More specifically, AI can learn from examples without relying on program-defined rules to figure out complex associations from multimodal image data beyond the current clinical practices, thus becoming a disruptive force in advancing intraoperative image guidance for treating SHD.

## 6. Conclusions

This paper reviews multimodal medical imaging modalities, the development of medical image fusion techniques, and recent advances in SHD treatment based on TEE fusion. It offers some perspectives for future research in this area.

With the advent of minimally invasive and percutaneous interventions in treating complex SHD, it is essential to consider live multimodal integration with advanced visualization systems as the platform for such interventions. With the potential to provide lower radiation doses and treatment times, TEE fusion imaging provides an enhanced visualization of cardiac anatomy for interventional procedures. New fusion techniques, like deep learning, can better align TEE with other images for real-time catheter guidance. Further research can test these AI-based fusion methods with TEE-CT and TEE-fluoroscopy imaging in a clinical setting. Furthermore, TEE-CT fusion can be introduced to real-time interventions following the completion of more research on their deep learning fusion.

## Figures and Tables

**Figure 1 diagnostics-13-02981-f001:**
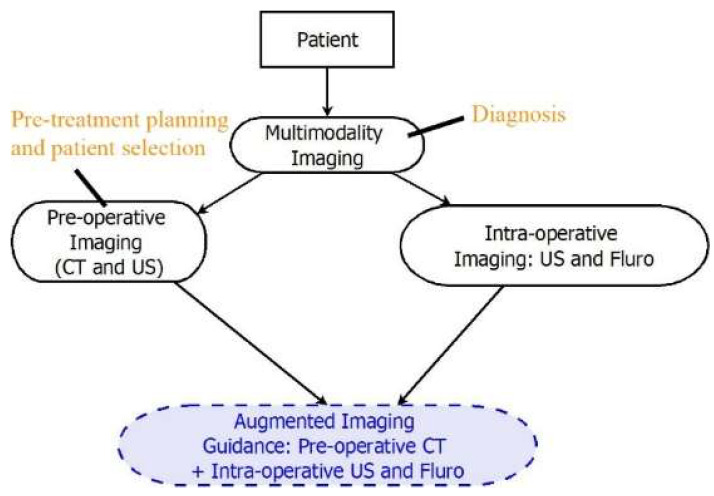
An illustration showing the critical role of imaging in SHD procedures. In the plot, CT, US, and Fluro are computed tomography, ultrasound (also known as echocardiography for cardiac applications), and fluoroscopy, respectively.

**Figure 2 diagnostics-13-02981-f002:**
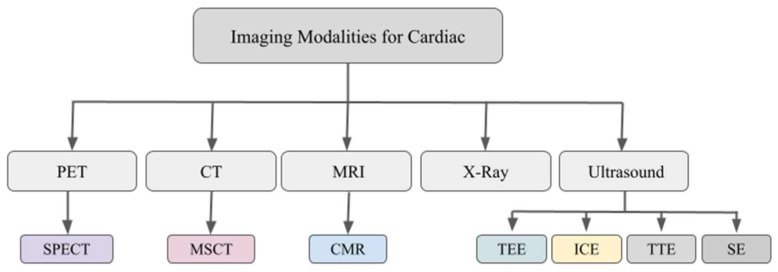
A diagram showing the imaging modalities frequently used for cardiac applications (PET—positron emission tomography, CT—computed tomography, MRI—magnetic resonance imaging, SPECT—single-photon emission computed tomography, MSCT—multi-spiral (or multi-slice) computed tomography, CMR—cardiovascular magnetic resonance, TEE—transesophageal echocardiography, ICE—intracardiac echocardiography, TTE—transthoracic echocardiography, and SE—strain elastography).

**Figure 3 diagnostics-13-02981-f003:**
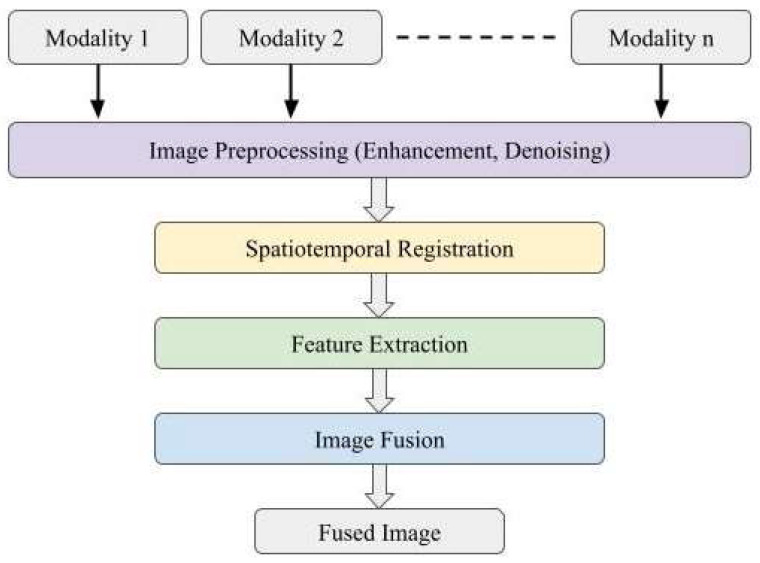
A schematic flowchart showing the steps in the multimodal image integration process.

**Figure 4 diagnostics-13-02981-f004:**
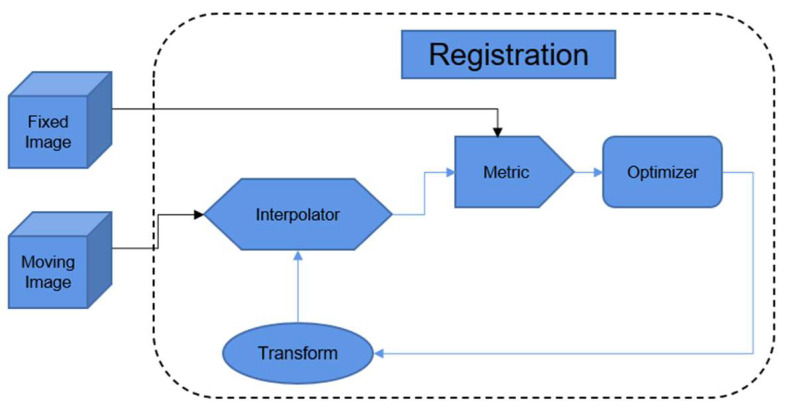
An illustrative diagram showing the procedures related to traditional image registration methods.

**Figure 5 diagnostics-13-02981-f005:**
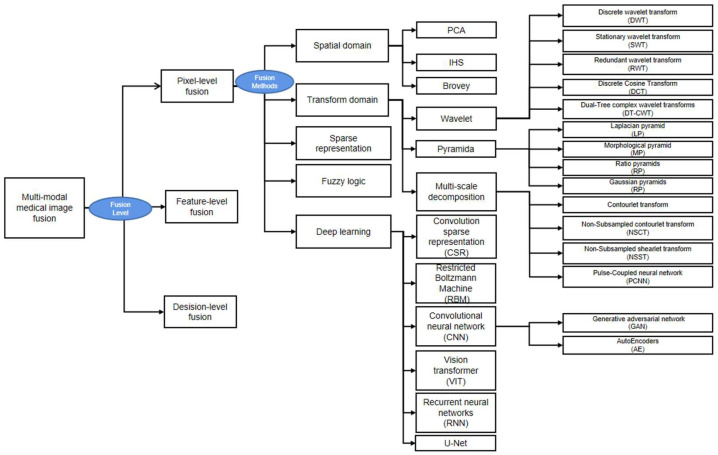
A diagram showing the various multimodal medical image fusion techniques.

**Figure 6 diagnostics-13-02981-f006:**
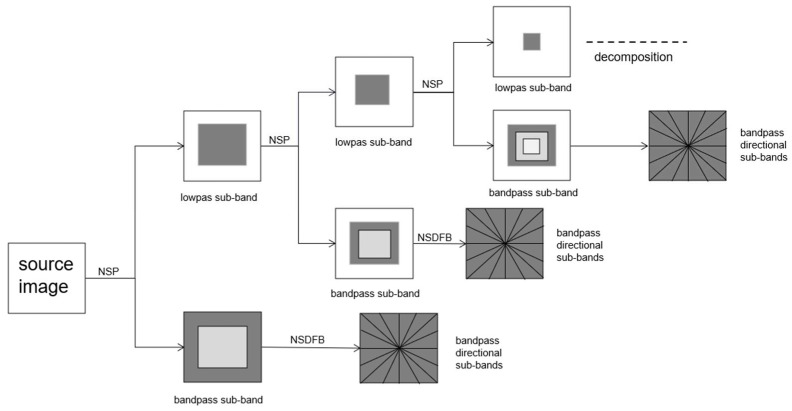
Decomposition process of NSCT.

**Figure 7 diagnostics-13-02981-f007:**
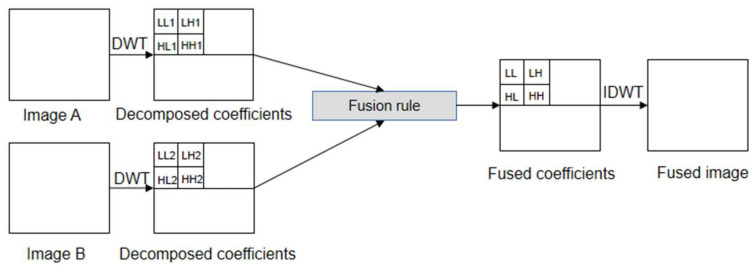
An illustration showing DWT-based image fusion.

**Figure 8 diagnostics-13-02981-f008:**
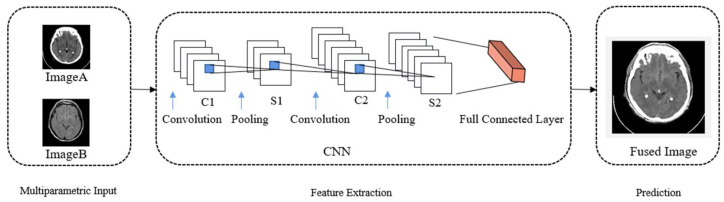
A general framework of deep-learning-based image fusion based on CNN.

**Figure 9 diagnostics-13-02981-f009:**
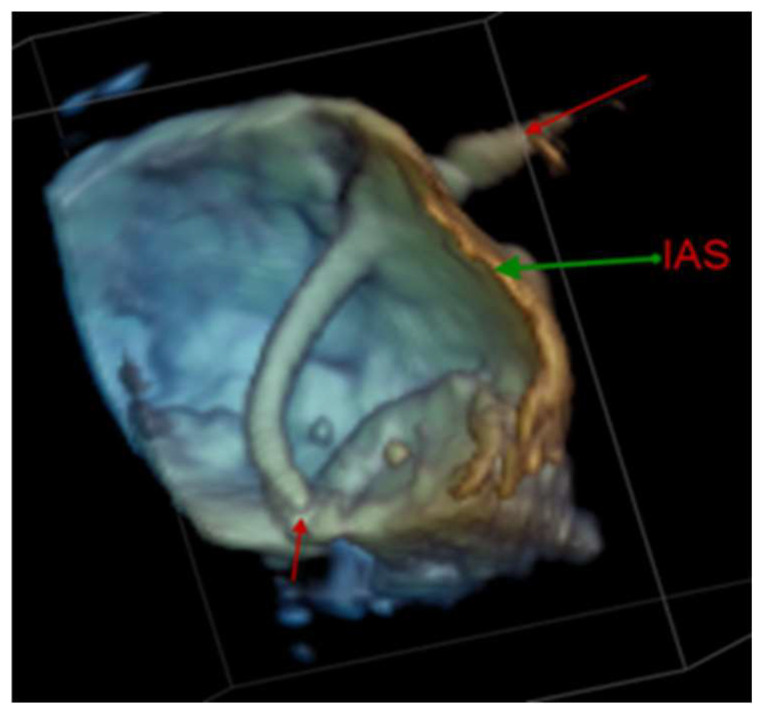
3D transesophageal echocardiography demonstrating a transseptal puncture from the left atrial perspective.

**Figure 10 diagnostics-13-02981-f010:**
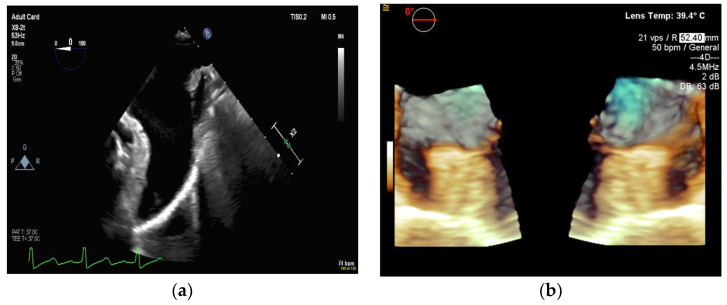
3D transesophageal echocardiography that demonstrates both sides of the left atrial appendage in profile: (**a**) an echocardiographic view and (**b**) a 3D reconstructed view.

**Figure 11 diagnostics-13-02981-f011:**
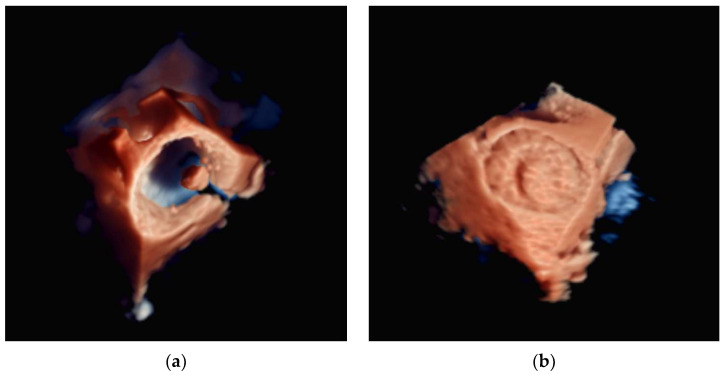
(**a**) 3D transesophageal echocardiography that demonstrates the mouth of the left atrial appendage. (**b**) A deployed left atrial appendage closure device.

**Figure 12 diagnostics-13-02981-f012:**
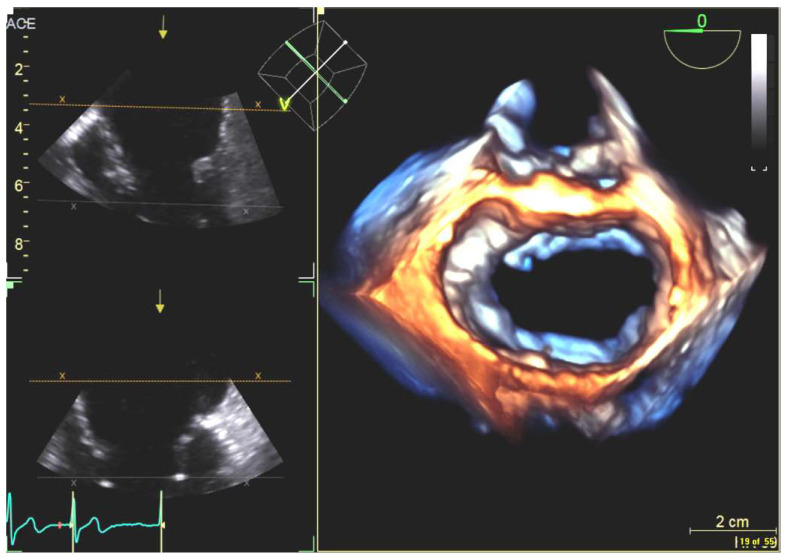
The use of 3D transesophageal echocardiography (TEE) in demonstrating the mitral valve from the left atrial perspective: (**left**) TEE imaging planes of the mitral valve and (**right**) the 3D volume.

**Figure 13 diagnostics-13-02981-f013:**
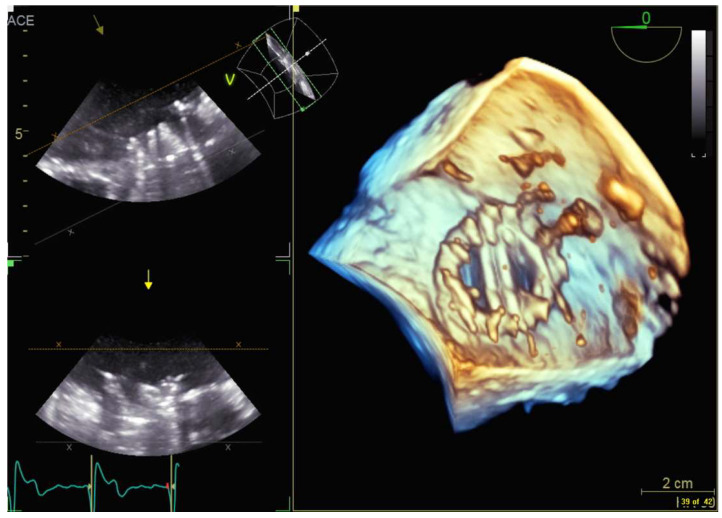
The use of 3D transesophageal echocardiography (TEE) in demonstrating a mitral valve replacement (MVR) from the left atrial perspective. The 2D TEE imaging planes of the MVR are shown to the left of the 3D volume.

**Figure 14 diagnostics-13-02981-f014:**
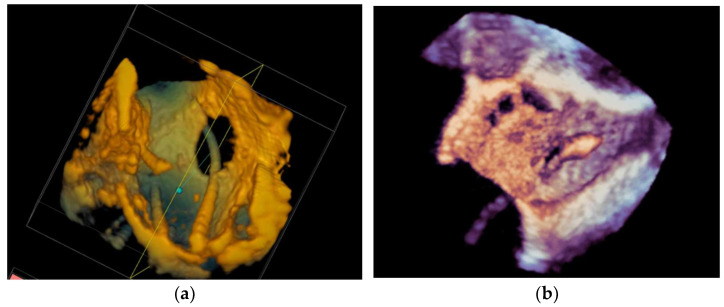
Variation in atrial septal defect morphology as illustrated by 3D TEE. Two catheters crossing the atrial septal defects—(**a**) one medium-sized and (**b**) one tiny defect.

**Figure 15 diagnostics-13-02981-f015:**
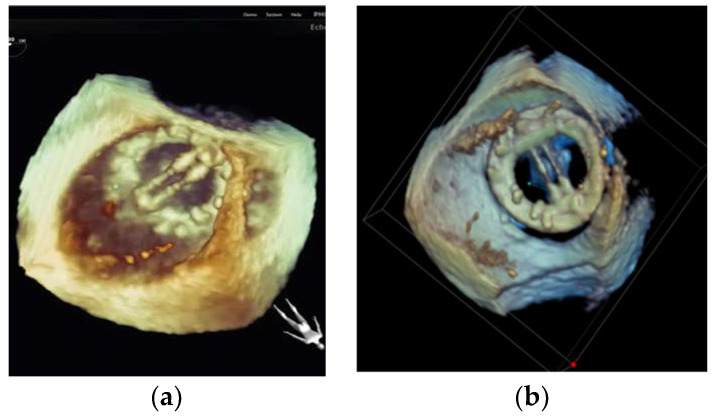
The 3D transesophageal echocardiography showing a paravalvular defect and successful closure with a device: (**a**,**b**) two 3D reconstructed TEE images showing defects.

**Figure 16 diagnostics-13-02981-f016:**
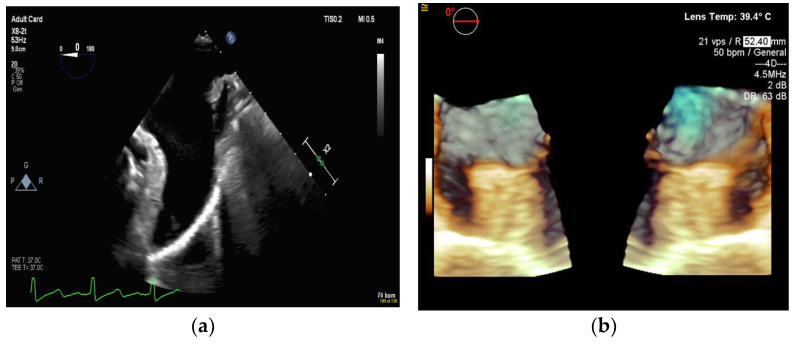
The 2D transesophageal echocardiography in demonstrating the deployed transcatheter aortic valve replacement: (**a**–**c**) multi-view and the 3D reconstructed aortic valve of interest.

**Figure 17 diagnostics-13-02981-f017:**
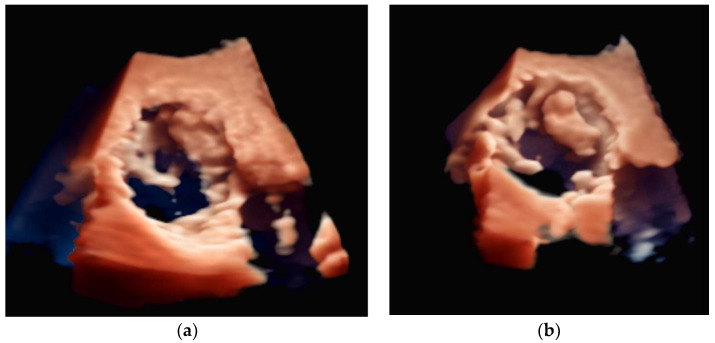
The 3D transesophageal echocardiography in demonstrating the deployed transcatheter aortic valve replacement with endocarditis on follow-up: (**a**,**b**) the reconstructed aortic valve defects from the 3D TEE data from different viewing angles.

## Data Availability

All relevant data have already been provided in the paper.

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
