# Peer review of "Advances in TEE-Centric Intraprocedural Multimodal Image Guidance for Congenital and Structural Heart Disease"

_diagnostics, 2023, doi:10.3390/diagnostics13182981_

Round 1
Reviewer 1 Report
The review is well written and very interesting for clinical cardiologists.
It accurately describes the advantages and limitations of transesophageal echocardiography (TEE) in intraprocedural guidance of several percutaneous structural interventions.
I have only one suggestion with regards to the limitations of TEE.
In the paragraph 2.4. Echoccardiography, after the lines 178-181, the Authors could add that during the last few years TEE has been considered a high-risk procedure for the possible transmission of highly contagious respiratory infections, such as COVID-19 (Please cite the following reference: PMID: 32503703). Accordingly, during the COVID-19 pandemic, some Authors have validated alternative noninvasive echocardiographic approaches, such as speckle tracking analysis of left atrial (LA) deformation properties. Impaired LA strain may largely predict left atrial appendage dysfunction and thrombosis by TEE in atrial fibrillation patients (Please cite the following reference: PMID: 34537932).
Reviewer 2 Report
The authors aim to present the role of TEE in the intraprocedural guidance of percutaneous structural interventions. In addition, the literature on multi-modal image integration has been reviewed in the paper.
The article summarizes the literature well. However, it would be good to present the literature summary in a more conspicuous way. I recommend that you summarize the literature with a table. (article year, method, the paper's main theme, image type (MRI, CT), performance parameter, etc.)
